# Impact of COVID-19 Vaccination on Pregnant Women

**DOI:** 10.3390/pathogens12030431

**Published:** 2023-03-09

**Authors:** Ishaan Chaubey, Harini Vijay, Sakthivel Govindaraj, Hemalatha Babu, Narayanaiah Cheedarla, Esaki M. Shankar, Ramachandran Vignesh, Vijayakumar Velu

**Affiliations:** 1Molecular and Cellular Biology, Department of Biology, Vanderbilt University, Nashville, TN 37235, USA; 2College of Arts and Sciences, University of Washington, Seattle, WA 98195, USA; 3Division of Microbiology and Immunology, Emory Vaccine Center, Emory National Primate Research Center, Emory University, Atlanta, GA 30329, USA; 4Laboratory Medicine, Department of Pathology, School of Medicine, Emory University, Atlanta, GA 30322, USA; 5Infection and Inflammation, Department of Biotechnology, Central University of Tamil Nadu, Thiruvarur 610005, India; 6Preclinical Department, Faculty of Medicine, Royal College of Medicine Perak, Universiti Kuala Lumpur, Ipoh 30405, Malaysia

**Keywords:** COVID-19 vaccination in pregnant women, cord blood, maternal and cord immune response

## Abstract

In light of the COVID-19 pandemic, researchers across the world hastened to develop vaccines that would aid in bolstering herd immunity. Utilizing mRNA coding and viral vector technology, the currently approved vaccines were required to undergo extensive testing to confirm their safety for mass usage in the general population. However, clinical trials failed to test the safety and efficacy of the COVID-19 vaccines in groups with weakened immune systems, especially pregnant women. Lack of information on the effects of vaccinations in pregnancy and the safety of fetuses are among the topmost reasons preventing pregnant women from receiving immunization. Thus, the lack of data examining the effects of COVID-19 vaccinations on pregnant women must be addressed. This review focused on the safety and efficacy of the approved COVID-19 vaccinations in pregnancy and their impact on both maternal and fetal immune responses. For that, we took the approach of combined systematic review/meta-analysis and compiled the available data from the original literature from PubMed, Web of Science, EMBASE and Medline databases. All articles analyzed presented no adverse effects of vaccination in pregnancy, with varying conclusions on the degree of effectiveness. The majority of the findings described robust immune responses in vaccinated pregnant women, successful transplacental antibody transfer, and implications for neonatal immunity. Hence, findings from the cumulative data available can be helpful in achieving COVID-19 herd immunization, including pregnant women.

## 1. Introduction

Beginning in December 2019, multiple cases of pneumonia were reported in the city of Wuhan, China, but further scans of the respiratory tract and the genetic sequencing of infected individuals revealed the presence of a novel coronavirus, later designated as SARS-CoV-2 [1]; the subsequent disease, now widely known as COVID-19. As the virus rapidly spread across the world, the World Health Organization (WHO) declared a global pandemic emergency, but many researchers saw a silver lining since it was found that the SARS-CoV-2 had a lower mortality rate as compared to SARS-CoV (the cause of the SARS outbreak in 2003) and MERS-CoV [2,3]. However, SARS-CoV-2 infection resulted in over 6.5 million deaths and 629 million infected cases worldwide (Center for Systems Science and Engineering [CSSE] at Johns Hopkins University [JHU], Baltimore, USA, 2021) and continued to infect further with evolving mutations.

The severity of the pandemic is universally acknowledged, with preventative measures in place, including social distancing, masking, maintenance of proper hygiene, and the daily monitoring of health conditions (Center for Disease Control and Prevention [CDC], Atlanta, USA, 2021). As the pandemic surged throughout the world, scientists pursued several vaccines that would aid in bolstering herd immunity and lowering the risk of viral infection [4,5]. Four main vaccines—Comirnaty, Spikevax, Evusheld, and Janssen produced by Pfizer, Moderna, Oxford-AstraZeneca, and Johnson & Johnson (J & J), respectively—were made available for public use in early to mid-2021. While the majority of the population seized the opportunity to become vaccinated, some groups remained hesitant, including pregnant women [6]. In a study conducted by the CDC from 14 December 2020 to 8 May 2021, only 11.1% of nearly 136,000 pregnant women had completed COVID-19 vaccination [7]. Because pregnant women were not included in the COVID-19 vaccine clinical trials, and there is very limited data on the safety and efficacy of the vaccine mechanisms during pregnancy [8]. Developmental and reproductive toxicology (DART) studies examining the effects of approval-pending vaccines on the entire range of the reproductive system in animals—have been conducted for the Pfizer and Moderna vaccines [9,10]. However, countries having access to less tested vaccines, such as AstraZeneca, has shown to have increased hesitation for immunization against SARS-CoV-2 amongst pregnant women population. On the other hand, studies show that pregnant women with prior comorbidities, such as diabetes, are more susceptible to COVID-19 complications [11,12], and the further lack of research on the side effects of vaccines in these groups increases the aversion to the administration of COVID-19 vaccinations despite the higher risk. 

Subsequently, in order to restore a small degree of normalcy in daily life, it is important for as many people as possible to become vaccinated against the SARS-CoV-2 virus/COVID-19. 

As observed in all prior pandemics, herd immunity is an essential component in minimizing the spread of viral infection or any disease within the community and, at some point, even offering protection to unvaccinated people [4,5]. Unfortunately, it is noted that the public still does not fully comprehend the concept of herd immunity. Essentially, herd immunity occurs when a large portion of the population gains immunity to a particular disease, thus hindering its dissemination among the members of the community [5]. In general, herd immunity can be achieved either through vaccination or the natural infection of the majority of the population; thereby, infected or vaccinated individuals gain humoral as well as cellular immunity to combat future infections [4,5]. For deadly diseases such as COVID-19, obtaining herd immunity through natural infection is a setback as it leads to a significant number of deaths in a population [13]. Hence, the alternative is to vaccinate the general population, which currently serves as the only effective method to ensure a significant decrease in the spread of the disease without placing more lives at risk. 

## 2. Methodology

We have followed the PRISMA reporting guidelines, and the PRISMA flowchart depicting the literature selection process is provided by Mohar et al., 2009 [14]. This research utilized published original articles and qualitative case study analysis to systematically compile, review, and analyze literature regarding the safety and efficacy of approved COVID-19 vaccinations in pregnancy and their repercussions on maternal and fetal immune responses. Subsequently, a systematic literature search on the PubMed, Web of Science, EMBASE, and Medline databases was conducted, and only scientific published literature was considered for this research in order to establish conclusions from the most accurate findings. The eligibility of scientific literature was determined by the following inclusion criteria: (a) articles focused on the safety and efficacy of approved COVID-19 vaccines for pregnant women, (b) articles focused on consequences of approved COVID-19 vaccines on fetal and maternal immune responses, and (c) articles possessed accredited sources and were peer-reviewed prior to publication. In addition, since this study focused on SARS-CoV-2 vaccination in pregnant women across the globe, no language restriction was used. However, any literature that focused on other coronaviruses, such as SARS-CoV or MERS-CoV, was excluded from consideration for this study. 

The search algorithms utilized for this systematic review were the following: “COVID-19 vaccination in pregnancy”, “neonatal immune outcomes following maternal COVID-19 immunization”, “transplacental transfer of COVID-19 antibodies following maternal immunization”, “fetal immune effects to maternal COVID-19 immunization”, “consequences of COVID-19 vaccination in pregnancy”, “IgM and IgG antibodies in women with SARS-CoV-2”, and “IgM and IgG antibodies in pregnant women with SARS-CoV-2”. Subsequently, these key search algorithms compiled a collection of 41 pieces of literature for understanding the safety and efficacy of approved COVID-19 vaccines for pregnant women and providing the capability to gain greater insight into the maternal and fetal immune responses after COVID-19 vaccination.

This review is mainly focused on approved vaccines such as the mRNA vaccine and the viral vector (adenoviral) vaccine and their mechanisms in an important group of people in the community, such as pregnant women, concerning the safety of both the mother and the child. One of the most prominent concerns of pregnant women is the lack of information on potential adverse side effects that are imposed by the COVID-19 vaccines on the fetus before birth and the neonate post-birth [6,8]. Therefore, we aimed to compile and analyze the so-far published data from experiments, clinical trials and literature reviews on the significance of SARS-CoV-2 immunity, in particular, the impact of vaccine-induced antibodies in the development of fetal immune systems. Furthermore, we also investigated the safety and efficacy of the various COVID-19 vaccines in pregnant women and their children, including a prolonged immune period in neonates. We focused on the safety and efficacy of the approved COVID-19 vaccinations in pregnancy and their impact on both maternal and fetal immune responses. For that, we took the approach of combined systematic review/meta-analysis and compiled the available data from the original literature from PubMed, Web of Science, EMBASE, and Medline databases. All articles analyzed presented no adverse effects of vaccination in pregnancy, with varying conclusions on the degree of effectiveness. The majority of the findings described robust immune responses in vaccinated pregnant women, successful transplacental antibody transfer, and implications for neonatal immunity. Hence, findings from the cumulative data available can be helpful in achieving COVID-19 herd immunization, including pregnant women. Finally, we emphasized the future contributions to increase knowledge on the side effects of COVID-19 vaccinations in fetal development that can help pregnant women be informed and decision making upon receiving COVID-19 vaccinations (Table 1).

## 3. Precedent for COVID-19 Vaccination in Pregnancy

Using precedent is an integral part of identifying the safety of a vaccine or any other drug when time and resources are too little to gather actual conclusive data. When considering the efficacy of vaccination in pregnancy in general, it is important to note that vaccines for illnesses such as tetanus, pertussis, and influenza are all administered during the second or third trimester of pregnancy [32]. One of the most well-known precedents for the safety of vaccination in pregnancy is shown in the influenza vaccine. The first clinical study for the influenza vaccine conducted during 2004–2005 in pregnant women investigated the effects of the vaccine and reported that pregnant women who were administered the influenza vaccine were 36% less likely to present symptoms of the flu itself [33]. Additionally, the study found that neonates of vaccinated mothers were 63% less at risk of influenza [33], implying a positive fetal immune response post-vaccination. While this data is not identical to the mechanisms of the approved COVID-19 vaccinations, it serves as a precedent supportive of the beneficial effects of vaccination in pregnancy for both mother and child. It is important to acknowledge that the current mRNA-based COVID-19 vaccines are the first mRNA vaccines to be tested on humans in large-scale phase-three clinical trials [34]. Since there is no vaccine precedent on specifically mRNA vaccines, it is crucial to examine other sources of data to come to an accurate and effective conclusion on the safety of the approved COVID-19 vaccinations in pregnancy. Adenoviral vaccines against the Ebola virus, namely Zabdeno (Ad26.ZEBOV) and Mvabea (MVA-BN-Filo), were the first to be granted marketing authorization by European Medicines Agency (WHO; Geneva, Switzerland; https://www.who.int/news-room/questions-and-answers/item/ebola-vaccines, accessed on 6 March 2023). The current adenoviral COVID-19 vaccinations are among the first to be used commercially in humans, although they have been in clinical trials for the past three decades [34]. Adenovirus vaccines currently undergoing clinical trials are used for illnesses such as influenza, tuberculosis, HIV, and Ebola [34]. A previous study conducted on the conferred maternal-fetal immune protection in mice against the Zika virus upon receipt of adenovirus vector-based vaccines concluded that adenovirus vaccines offer robust protection against the Zika virus in pregnant mice. The vaccine-elicited antibodies provide neonatal protection as well [31,35]. Although this study does not serve as a human vaccination precedent, it is still an important conclusion in identifying the potential benefits of adenovirus vaccinations against COVID-19 in pregnant women. 

## 4. V-Safe Findings of COVID-19 Vaccination 

The effectiveness of the COVID-19 vaccines on pregnant women, as well as fetuses and neonates, were recorded in v-safe, a voluntary registry set by the CDC following approval for administration of the COVID-19 vaccines where pregnant women can log their side effects to the COVID-19 vaccinations following administration [21]. A study conducted by the CDC themselves analyzed the effects of the COVID-19 vaccinations in registered v-safe people who received at least one dose of the mRNA-based vaccine preconception or after 20 weeks of gestation, aiming to identify the risk of spontaneous abortions (SABs)—defined as miscarriages occurring from 6–20 weeks of gestation. Among the 2500 participants in this study, the risk of SABs was seen to be 14.1%, with a standardized risk of 12.8%, indicating that the mRNA COVID-19 vaccines are not associated with an increased risk of miscarriages [16]. However, the limitations of this observational study included a lack of a control group with unvaccinated pregnant women, a homogenous study group in regards to racial and ethnic groups, self-reported data which may have biased results due to participants’ heightened anxiety about miscarriages, and possible enrollment bias due to the voluntary nature of the registry. Additionally, another study conducted among 3958 pregnant women enrolled in the v-safe registry concluded that 86.1% of pregnancies resulted in live births, 13.9% in pregnancy losses, 9.4% in preterm births, and 3.2% in small gestational sizes of the neonate [21]. The adverse effects in pregnancies following the administration of the COVID-19 vaccines presented no jarring discrepancies from pregnancies prior to the pandemic itself, indicating that percentages were as expected and that there were no adverse effects of the COVID-19 vaccinations. It is important to note, however, that the majority of the participants received the vaccinations later in their pregnancy, so a follow-up observational study must be conducted on the effects of vaccination earlier in pregnancy. 

## 5. DART Findings of COVID-19 Vaccination

Developmental and reproductive toxicology (DART) studies examine the effects of medication and vaccines in the entire reproductive system of animals and are extremely crucial in hypothesizing the effects of those medications in human reproductive systems [6]. Consequently, drawing upon the findings of DART studies conducted with the COVID-19 vaccinations contributes to summarizing the effects of the vaccinations on pregnant women. Rats were used as a model organism in a DART study examining the effects of the mRNA Pfizer vaccination on pregnancy and represented one of the first published data evaluating the full extent of the mechanisms of the mRNA vaccine [9]. Results concluded that there were no adverse effects on fetal or neonatal survival and development, and robust neutralizing antibodies were recorded both during gestation and lactation, indicating that neonates were given prolonged immunity after birth [9]. Additionally, summarized data by the World Health Organization (WHO) concluded that DART studies conducted with the AstraZeneca adenovirus vaccines likewise present no adverse vaccine-associated risks during pregnancy (World Health Organization [WHO], 2021). 

## 6. Fetal Immune Mechanisms Post-Vaccination

The fetal immune mechanisms must first be understood to better comprehend the prolonged immunity of neonates post-vaccination. For instance, it is widely known that passive immunization allows for the transplacental passage of antibodies into fetal cord circulation post-maternal infection [36,37] or vaccination, so it can thus be inferred that maternal vaccination both protects the mother and the fetus [11]. However, it is still uncertain whether breast milk IgG transfer provides neonatal protection. Thus, it is important to understand the full extent of passive immunity to draw a complete conclusion on the long-term effects of antibody titers in neonates post-vaccination. 

A study aimed at identifying specific maternal and cord antibody titers against SARS-CoV-2 following Pfizer vaccination identified high anti-S, or anti-spike protein, antibodies in the cord blood after birth, indicating that maternal immunization may have been provided through transplacental antibody transfer [26]. This study also identified a correlation between the time from vaccination to delivery and antibody transfer, which aids in providing guidance on the most beneficial time in pregnancy for people to receive a vaccination. It was also found that SARS-CoV-2-specific IgG antibodies can be detected in cord blood following the first dose of the mRNA Moderna vaccination [38]. Overall, it is observed that the presence and transfer of maternal antibodies post-mRNA COVID-19 vaccination in cord blood, concluding that further studies will be needed to quantify antibody titers, including neutralizing potency. Outcomes of the research, however, were hopeful, implying successful and effective SARS-CoV-2 specific antibody transfer in cord blood to protect against neonatal complications. 

## 7. Effects of COVID-19 Vaccines on Neonatal Viral Immunity

While it is crucial to understand the immediate effects of the COVID-19 vaccinations on fetuses, as they are in a much more vulnerable state, it is likewise important to identify whether the COVID-19 vaccinations will benefit the neonate post-birth. This review also defines neonatal immunity to be the successful transfer of antibodies through the placenta (identified by antibody presence in cord blood) and/or breast milk. Such an investigation summarizes the entire scope of the effectiveness of the COVID-19 vaccines since any positive or negative outcomes will be fully determined as neonates no longer rely on immune protection from the mother, with the exception of lactation. Identifying the prolonged viral immunity in neonates alleviates concerns in pregnant mothers on the long-term effects of COVID-19 vaccination on neonates. Subsequently, this greater insight into the duration of innate immunity in neonates increases pregnant women’s inclination to receive COVID-19 vaccinations. As the conditions behind this scope of research are either too recent or still developing, evidence is varied and largely serves purely as background for further studies. However, the usage of precedent serves useful in addressing the concern of neonatal effects, as the past Zika adenovirus vaccination DART study conducted on rats concluded that pups born to vaccinated mothers were protected against Zika challenges post-birth, and was assumed to be a result of passive immunity [39]. Though further research is required to identify the extent of this conclusion on humans, as well as its validity, the DART study is a major starting point for easing hesitancy in concerned pregnant patients. Additionally, a recent study found that mRNA vaccines are highly effective in the robust production of SARS-CoV-2-specific antibodies in pregnant women [31,40,41,42]. Vaccine-induced titers were found to be equivalent in pregnant, lactating, and non-pregnant women, suggesting that COVID-19 vaccines transfer antibodies both transplacental and through breast milk and provide context to the mechanisms behind which neonates can secure prolonged immunity. These new findings can be presented as tentative data for additional support for the acceptance of COVID-19 vaccinations in pregnant women [31,41,42]. 

Summatively, various literature and studies indicate the lack of adverse effects of both mRNA and adenovirus vaccines in fetal and neonatal immune development, but further research is required to reach a conclusive statement. However, analyzed studies have shown more beneficial effects of the COVID-19 vaccines, which tentatively serve to be an affirming and convincing factor in confirming the safety and efficacy of the COVID-19 vaccines in pregnant mothers as well as the prolonged immunity of neonates. Furthermore, since most of the scientific literature signifies the strong unlikelihood of negative consequences after the administration of the COVID-19 vaccines, pregnant women are subsequently more encouraged to be vaccinated, thus enhancing global efforts in reaching the herd immunity threshold against SARS-CoV-2 infection. As indicated in Table 1, all analyzed articles concluded the lack of adverse effects, but it should be noted that some studies primarily focused on both immunological and physiological outcomes in order to verify the safety of maternal COVID-19 immunization. Investigations of maternal and fetal outcomes came to one common consensus: Maternal COVID-19 immunization does not result in negative outcomes for both the mother and child [15,16,17,21,22,24,25,28], as supported by Table 1. Though not all articles focused specifically on identifying the possibility of adverse effects, all literature nonetheless concluded that the COVID-19 vaccines in pregnancy produced only beneficial outcomes so far. Studies investigating neonatal immunity concluded the presence of antibodies in cord blood samples taken from children of mothers who received the COVID-19 vaccinations, indicating the success of transplacental antibody transfer. Furthermore, some studies included an additional objective of examining and affirming the successful transfer of antibodies through breast milk (Table 1; [8,15,18,20,22,28,29].

The degree of antibody transfer was broken down by findings identifying the robust transfer of IgG antibodies in general, RBD-IgG antibodies, and anti-S antibodies, either via the placenta or breast milk. Several studies found that there was a robust antibody transfer of IgG antibodies since these antibodies were detected in cord blood samples and breast milk (Table 1; [8,18,19,20,23,26,29]. Additionally, studies also examined the ratios of certain IgG antibodies in mother-child dyads, specifically, RBD-IgG antibodies and concluded their robust transfer (Table 1; [8,18,20,23,26,27,29]. Finally, it should be noted that some of the analyzed studies did not specify their conclusions on IgG transfer, but there were a few studies that showed the robust transfer of anti-S antibodies [8,19,22,23,26,29,30]. Due to the nature of the studies conducted by [15,21], identification of successfully transferred antibodies was not provided. However, both studies did conclude that a robust transfer did occur. Lastly, the analyzed literature provided suggestions for investigating an optimal vaccination time frame in order to maximize antibody transfer from mother to child. For instance, Ref. [19] emphasized the need for strict adherence to a vaccination schedule for pregnant women in order to maximize the observed beneficial effects of immunization. (Table 1). Additionally, Rottenstreich et al. (2022) identified a peak in the robust production and transfer of IgG antibodies when pregnant women were immunized in their early third trimester (Table 1). 

## 8. Conclusions

The majority of the studies conducted so far were focused on the safety of mRNA biotechnology rather than including the effects of adenovirus-based COVID-19 vaccines. However, the impacts of adenoviral technology in pregnancy must also be investigated in order to inform the pregnant population in countries that either have a surplus of adenoviral COVID-19 vaccines or have not yet approved mRNA-based vaccines. Additionally, this review analyzed only two articles with implications for optimal vaccination timeframes. Future investigations should prioritize exploring the ideal time in pregnancy for mothers to receive immunization in order for both mother and child to maximize their immune responses. Furthermore, we focused on investigating the safety and efficacy of the COVID-19 vaccines in pregnancy in order to contribute to the scarce amount of information present for pregnant mothers to weigh the benefits and risks of maternal immunization, especially considering the vaccine methods that have not been administered for general use prior to the COVID-19 pandemic. Through an in-depth analysis of existing literature examining the effects of the two approved vaccine methods on fetal development and implications of neonatal immunity, as well as a combined systematic review/meta-analysis approach characterizing and comparing the immune responses between vaccinated and unvaccinated mother: child dyads, we conclude that there is a lack of negative outcomes associated with maternal COVID-19 immunization. Moreover, studies supported the postulation that vaccinated pregnant women may produce a more robust antibody response in comparison to unvaccinated pregnant women when they get infected. Substantial evidence also concluded that significant transplacental and lactational antibody transfer to the child offers protection even if they are exposed to the disease despite effective care. However, accurate and conclusive evidence noting the safety and efficacy of the COVID-19 vaccines in pregnancy will serve as a crucial factor in alleviating the decision-making process of pregnant women. More specifically, additional information in regard to supporting beneficial outcomes in the fetus as well as immune protection for the neonates will resolve hesitancy towards administering COVID-19 vaccination in the pregnant women population [6]. Overall, immunization of the pregnant women population is considered one of the best strategies that will help fight against the pandemic and take us one step closer to achieving herd immunity and consequently regaining a sense of normalcy in daily life. 

## Figures and Tables

**Table 1 pathogens-12-00431-t001:** Impact of COVID-19 Vaccination in pregnant women.

Country	Adverse Effects	Vaccine Company	Vaccine	Article Description	Findings	Article
Israel	No adverse effects reported	Pfizer BioNTech	mRNA	Efficient transfer of neutralizing antibodies from mother to cord (Pfizer vaccine)	Results proved rapid rise in antibodies (specifically IgG and spike-specific IgG) + their effective transfer across placenta to neonate in vaccinated pregnant women; Reported Pfizer vaccine efficacy: 94%	(Beharier et al., 2021) [8]
Israel; USA	No adverse effects reported	Pfizer BioNTech	mRNA	Observational cohort study of vaccinated and unvaccinated pregnant people determining percent efficacy of the Pfizer mRNA vaccine in documented, symptomatic, and hospitalized COVID-19 groups	High vaccine effectiveness in pregnant women; Reported Pfizer vaccine efficacy: 96% against documented infection and 97% against symptomatic infection after receipt of second vaccine dose	(Dagan et al., 2021) [15]
USA	No adverse effects reported	Pfizer BioNTech	mRNA	Observational study conducted to assess the risk of spontaneous abortions (miscarriages) preconception and during first 20 weeks of gestation following receipt of specifically mRNA COVID-19 vaccines	Resulted in 12.8% risk of spontaneous abortions (SABs) indicating that there are no adverse effects of mRNA vaccination in pregnant people or the health of the fetus	(Zauche et al., 2021) [16]
Argentina; USA; UK	No adverse effects reported	Pfizer BioNTech + Moderna	mRNA	Rapid study conducted to identify the adverse effects of COVAX approved COVID-19 vaccinations on pregnant people to determine whether its administration is safe	Concluded that COVAX approved COVID-19 vaccinations are safe for use in pregnancy and pose no concerns to fetal + neonatal life	(Ciapponi et al., 2021) [17]
USA	No adverse effects reported	Moderna + Pfizer BioNTech	mRNA	Assessed immunogenicity of approved COVID-19 mRNA vaccines in pregnant + lactating women	RBD + neutralizing antibody titers higher in vaccinated groups opposed to pre vaccination; Antibodies also identified in infant cord blood indicating transplacental antibody transfer; Antibodies identified in breast milk indicating newborns protected by vaccination	(Collier et al., 2021) [18]
USA	No adverse effects reported	Moderna + Pfizer BioNTech	mRNA	Aimed to identify whether physiological + hormonal changes make pregnant patients less responsive to vaccination or induce altered immune responses	Findings suggest pregnancy promotes resistance to generating pro-inflammatory antibodies; Provides evidence that pregnant women must adhere to a strict booster vaccine schedule in order to fully mature immune response	(Atyeo et al., 2021) [19]
Israel	None found	Pfizer BioNTech	mRNA	Aimed to identify transplacental transfer of COVID-19 antibodies in vaccinated pregnant women	Concluded efficient transplacental transfer of COVID-19 IgG antibodies in pregnant women vaccinated w/ Pfizer; Also noted implications for neonatal humoral immunity	(Nir et al., 2021) [20]
USA	No adverse effects reported	Moderna + Pfizer BioNTech	mRNA	Used vaccine monitoring systems to characterize safety of mRNA COVID-19 vaccination in third trimester	No adverse effects noted; Further monitoring required to assess total outcomes following maternal vaccination	(Shimabukuro et al., 2021) [21]
UK	No adverse effects reported	AstraZeneca	Adenovirus	Nonclinical DART study to evaluate effects of Astrazeneca on fertility + reproductive system of mice + postnatal outcomes	No adverse effects; Robust immune responses in mothers and pups; Transplacental and lactational transfer of antibodies	(Stebbings et al., 2021) [22]
Israel	No adverse effects reported	Pfizer BioNTech	mRNA	Provided assessment of impact of COVID vaccination on transplacental transfer during early or late third-trimester vaccination	Vaccination in early third trimester enhances neonatal protection; Early third-trimester immunization yields higher neonatal antibody concentrations (specifically IgG types)	(Rottenstreich et al., 2022) [23]
USA	No adverse effects reported	Pfizer BioNTech + Moderna	mRNA	Described maternal, neonatal, + obstetrical outcomes of women who received mRNA vaccinations	Noted no adverse effects similar to results of other literature	(Trostle et al., 2021) [24]
Israel	No adverse effects reported	Pfizer BioNTech	mRNA	Investigated association between Pfizer vaccine + pregnancy course/outcomes; Identified characteristics related to vaccination in pregnancy	Prenatal vaccination does not present adverse outcomes or newborn complications; Pfizer efficacy higher than 85% in regards to risk of symptomatic infection and transmission	(Wainstock et al., 2021) [25]
Poland	No adverse effects reported	Pfizer BioNTech	mRNA	Aimed to identify titers of specific maternal/cord antibodies against COVID-19 S-protein post antenatal vaccination along with ratio of umbilical cord + maternal antibody titers	High anti-S antibodies in cord blood suggest maternal immunization provides protection to newborns via transplacental antibody transfer; Significant cord serum antibody titer levels suggest optimal vaccination time	(Zdanowski & Wasniewski, 2021) [26]
Indonesia; USA	No adverse effects reported	Pfizer BioNTech + Moderna	mRNA	Provided greater insight into the efficacy of current COVID-19 vaccines with regards to maternal and neonatal antibody responses, transplacental transfer of antibodies, and any reported adverse effects in either the mother or infant	No severe complications reported; Very mild to moderate COVID symptoms observed after vaccination of pregnant women resulting in various local and systemic adverse events; Transplacental IgG antibody transfer of also observed when examining cord blood antibody levels; Reported Pfizer efficacy: 94.1%; Reported Moderna efficacy: 95%	(Pratama et al., 2022) [27]
USA	No adverse effects reported	Pfizer-BioNTech + Moderna	mRNA	Aimed to illustrate the effectiveness of 2-dose maternal COVID-19 vaccination during pregnancy and its subsequent implications for the developing neonate	Maternal antibodies detected in maternal sera at delivery along with breast milk and infant sera suggesting that maternal COVID-19 vaccination during pregnancy may protect infants aged 6 months or less from severe SARS-CoV-2 infection	(Halasa et al., 2022) [28]
USA	No adverse effects reported	Pfizer BioNTech + Moderna + Johnson & Johnson	mRNA	To provide greater insight into the maternal antibody response and transplacental antibody transfer through the umbilical cord with regards to the vaccine administered to the pregnant woman and the trimester of vaccination	Confirmed presence of maternal IgG and anti-S spike antibodies in the developing newborn following transplacental antibody transfer; Spike-specific antibodies significantly higher in concentration in cord blood after administering Moderna or Pfizer vaccines versus cord blood after J & J vaccine	(Atyeo et al., 2022) [29]
USA	No adverse effects reported	Pfizer BioNTech + Moderna + Johnson & Johnson	mRNA	Focused on characterizing the persistence of vaccine-induced anti-S IgG antibodies in infants born to vaccinated mothers with comparison to infants born to mothers with natural SARS-CoV-2 infection	High anti-S antibodies for 6-month old infants born to COVID-vaccinated mothers compared to infants born to mothers with natural SARS-CoV-2 infection; Findings suggest maternal immunization provides protection to newborns via transplacental antibody transfer	(Shook et al., 2022) [30]
USA	No adverse effects reported	Pfizer BioNTech + Moderna	mRNA	Focused on characterizing SARS-CoV-2 and COVID-19 vaccine-induced binding and neutralizing antibodies in maternal and cord blood	COVID-19 vaccinated pregnant women raised higher binding and neutralizing antibodies than in all SARS-CoV-2 infected pregnant women	(Dude et al., 2023) [31]

## Data Availability

The datasets generated and/or analyzed during the current study are available from the corresponding author upon reasonable request.

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
