# Peer review of "Impact of COVID-19 Vaccination on Pregnant Women"

_pathogens, 2023, doi:10.3390/pathogens12030431_

Round 1

Reviewer 1 Report

Line 24 and 34 – What population are we talking about? Where are there so many pregnant women that they have not been vaccinated to achieve herd immunity? This is a wishful conclusion - please change the whole narrative of the article or demonstrate that vaccinating pregnant women would contribute to herd immunity.

The failure to achieve herd immunity was primarily influenced by the following factors: the spread of fake news via social media, lack of trust in governments and reluctance to protect and vaccinate the main working population. The losing race of vaccination against the variability of the virus was also an important factor. Apart from the size of the population, pregnant women are the least migrating population, avoiding large groups of people, taking better care of themselves and not unnecessarily exposing themselves to infection.

Please consider this problem in terms of exclusion of this social group from available prevention, stress, uncertainty, difficulties in making a decision to get vaccinated and taking responsibility not only for yourself, and limited functioning.

Line 53-54 - This statement is incomplete. In the development of each vaccine, the aim is to obtain a product with maximum effectiveness, and only use in real conditions verifies this wishful goal. The researchers hoped to be effective enough to block transmissions, but that efficiency depends not only on the product but also on the rapid, simultaneous implantation of the majority of the population. Effectiveness at the level of not developing the disease, having a milder course or not being hospitalized is also a success. Vaccine developers know this.

Line 74 - either SARS-CoV-2 virus or COVID-19. The first option is probably better.

Line 100-105- this seems to be the guiding line of this article. Not gaining herd immunity.

Line 106 - much better to use medical nomenclature and replace "precedent" with "off-label use" almost everywhere? It will then be understandable for medics.

Line 119 - "precedent" or reference?

Line 125-128 - On May 29, 2020, Zabdeno (Ad26.ZEBOV) and Mvabea (MVA-BN-Filo) were approved for trading, so the COVID-19 adenoviruses are not the first.

The meta-analysis methodology was not given, the keywords that were used for the database review were not given - please complete.

Author Response

We are pleased to know that the reviewers reviewed our manuscript favorably, and we thank all the reviewers for their enthusiasm and encouraging comments regarding our data review on “the impact of COVID-19 vaccination in pregnant women”.  All reviewers agreed with minor revisions but they also had few concerns (listed below). However, these concerns are easily addressable. Below is our point-by point response.

Reviewer 1:

Line 24 and 34 – What population are we talking about? Where are there so many pregnant women that they have not been vaccinated to achieve herd immunity? This is a wishful conclusion - please change the whole narrative of the article or demonstrate that vaccinating pregnant women would contribute to herd immunity. The failure to achieve herd immunity was primarily influenced by the following factors: the spread of fake news via social media, lack of trust in governments and reluctance to protect and vaccinate the main working population. The losing race of vaccination against the variability of the virus was also an important factor. Apart from the size of the population, pregnant women are the least migrating population, avoiding large groups of people, taking better care of themselves and not unnecessarily exposing themselves to infection. Please consider this problem in terms of exclusion of this social group from available prevention, stress, uncertainty, difficulties in making a decision to get vaccinated and taking responsibility not only for yourself, and limited functioning.

We thank the reviewer for the encouraging comments and we addressed all the comments raised by the reviewer 1. Please see below point by point response for the comments.

Line 53-54 - This statement is incomplete. In the development of each vaccine, the aim is to obtain a product with maximum effectiveness, and only use in real conditions verifies this wishful goal. The researchers hoped to be effective enough to block transmissions, but that efficiency depends not only on the product but also on the rapid, simultaneous implantation of the majority of the population. Effectiveness at the level of not developing the disease, having a milder course or not being hospitalized is also a success. Vaccine developers know this.

Line 74 - either SARS-CoV-2 virus or COVID-19. The first option is probably better.

Thanks for the reviewer, appropriate changes have been made

Line 100-105- this seems to be the guiding line of this article. Not gaining herd immunity.

Appropriate changes have been made in the lines (108-110) as suggested by the Reviewer.

Line 106 - much better to use medical nomenclature and replace "precedent" with "off-label use" almost everywhere? It will then be understandable for medics.

For the sake of general audience, we would like to use the word "precedent" instead of "off-label use"

Line 119 - "precedent" or reference?

Precedent would fit good so we would like to stick to the word ‘Precedent’

Line 125-128 - On May 29, 2020, Zabdeno (Ad26.ZEBOV) and Mvabea (MVA-BN-Filo) were approved for trading, so the COVID-19 adenoviruses are not the first.

The corrections were made in the lines (129-134) as per the reviewer’s suggestion.

The meta-analysis methodology was not given, the keywords that were used for the database review were not given - please complete.

The corrections were made in the lines (129-134) as per the reviewer’s suggestion.

Reviewer 2 Report

the Ms pathogens 2228442 can be accepted with minor revisions only for several typos.

In my opinion, the work makes a significant contribution to knowledge of the effects of Covid-19, although there are already numerous works on the subject. The topic explored by the authors on the impact of vaccination on pregnant women and newborns confirms the observations and data provided by the pharmaceutical manufacturers of the vaccine. The scarce or no presence of serious side effects in prepartum and post-partum conditions can help convince other patients in the same conditions to face vaccination with serenity, increasing the percentage of individuals vaccinated in order to achieve the desired herd immunity from the first vaccination events, avoiding the disinformation campaign prompted by no-vaxes.

The manuscript is structured in a way that is understandable even to non-experts.

The bibliography is well-reported.

A thorough re-reading of the text is recommended, due to the presence of some typos.

Author Response

Reviewer 2:

the Ms pathogens 2228442 can be accepted with minor revisions only for several typos.

In my opinion, the work makes a significant contribution to knowledge of the effects of Covid-19, although there are already numerous works on the subject. The topic explored by the authors on the impact of vaccination on pregnant women and newborns confirms the observations and data provided by the pharmaceutical manufacturers of the vaccine. The scarce or no presence of serious side effects in prepartum and post-partum conditions can help convince other patients in the same conditions to face vaccination with serenity, increasing the percentage of individuals vaccinated in order to achieve the desired herd immunity from the first vaccination events, avoiding the disinformation campaign prompted by no-vaxes. The manuscript is structured in a way that is understandable even to non-experts. The bibliography is well-reported. A thorough re-reading of the text is recommended, due to the presence of some typos.

Thanks to the reviewer for the positive feedback on the significance and core objective of this review. A thorough reading of the text for typos was considered seriously as recommended by the reviewer and corrections have been made. Please see the revised manuscript with yellow highlighted.

Reviewer 3 Report

REVIEW REPORT

This research paper addressed the topmost important question of the safety and efficacy of the approved COVID-19 vaccinations in pregnancy and their impact on both maternal and fetal immune responses. The review was based on data from PubMed. This is part of an ongoing effort to bolster herd immunity so that vulnerable populations could be protected by a screening net of immune people.  This Systematic review is a compilation of evidence from various studies like DART, and V-Safe which found that vaccines did not cause spontaneous abortions (SABs) in human and animal studies. There is also a claim that trans-placental and breast milk-secreted antibodies against SARS-CoV-2 provide protective immunity to the baby. The review focused only on the mRNA vaccine and the viral vector (adenoviral) vaccine and did not include other vectors of delivery like Live attenuated, Sub-unit vaccine, Inactivated vaccines, and protein-based vaccines.

On one hand, the Systematic review included copious literature available on the subject and was scientifically accurate. They analyzed both human and animal studies which strengthens their claims. Finally, the latest literature has been consulted, so the conclusion could be assumed to be relevant also.

On the other hand, the Authors have mixed the conclusions of both types of vaccines into a single heterogeneous group. Even the studies included in their synthesis are a potpourri of races, economic conditions, stages of pregnancy so on without any stratification. The review compiled only 2 papers about implications for optimal vaccination timeframes which is fewer.

Author Response

Reviewer 3:

This research paper addressed the top most important question of the safety and efficacy of the approved COVID-19 vaccinations in pregnancy and their impact on both maternal and fetal immune responses. The review was based on data from PubMed. This is part of an ongoing effort to bolster herd immunity so that vulnerable populations could be protected by a screening net of immune people.  This Systematic review is a compilation of evidence from various studies like DART, and V-Safe which found that vaccines did not cause spontaneous abortions (SABs) in human and animal studies. There is also a claim that trans-placental and breast milk-secreted antibodies against SARS-CoV-2 provide protective immunity to the baby. The review focused only on the mRNA vaccine and the viral vector (adenoviral) vaccine and did not include other vectors of delivery like Live attenuated, Sub-unit vaccine, Inactivated vaccines, and protein-based vaccines.  On one hand, the Systematic review included copious literature available on the subject and was scientifically accurate. They analyzed both human and animal studies which strengthens their claims. Finally, the latest literature has been consulted, so the conclusion could be assumed to be relevant also.
On the other hand, the Authors have mixed the conclusions of both types of vaccines into a single heterogeneous group. Even the studies included in their synthesis are a potpourri of races, economic conditions, stages of pregnancy so on without any stratification. The review compiled only 2 papers about implications for optimal vaccination timeframes which is fewer.

We thank the reviewer for acknowledgement of the importance of this review on COVID-19 vaccination in pregnant women. As reviewer rightly mentioned, this review is aimed to spotlight the inclusion of pregnant women into the vaccinated population to meet the success of herd immunity to protect the vulnerable population. We are thankful that the reviewer assured the scientific accuracy of this review as presented based on the human and animal study data and the timeline relevant to the current scenario. Please see the revised manuscript with yellow highlighted.
